# ENHANCING ATTENTION WITH EXPLICIT PHRASAL ALIGNMENTS

## ABSTRACT

The attention mechanism is an indispensable component of any state-of-the-art neural machine translation system. However, existing attention methods are often token-based and ignore the importance of phrasal alignments, which are the backbone of phrase-based statistical machine translation. We propose a novel phrase-based attention method to model n-grams of tokens as the basic attention entities, and design multi-headed phrasal attentions within the Transformer architecture to perform token-to-token and token-to-phrase mappings. Our approach yields improvements in English-German, English-Russian and English-French translation tasks on the standard WMT'14 test set. Furthermore, our phrasal attention method shows improvements on the one-billion-word language modeling benchmark.

## 1   INTRODUCTION

The encoder-decoder neural architectures have established breakthroughs in many natural language processing (NLP) tasks including machine translation (Luong et al., 2015), summarization (See et al., 2017), and parsing (Vinyals et al., 2015). Particularly for machine translation, most state of the art neural machine translation (NMT) models possess attention mechanisms to perform alignments of the target tokens to the source tokens. The attention module therefore plays a role analogous to the word alignment model in Statistical Machine Translation or SMT (Koehn, 2010). In fact, the Transformer network introduced recently by Vaswani et al. (2017) achieves state-of-the-art performance in both speed and BLEU scores (Papineni et al., 2002) by using only attention modules. Furthermore, its proposed self-attention layer extends the application of attention way beyond the scope of machine translation and achieves tremendous success in other NLP tasks such as contextual representation learning (Devlin et al., 2018) and machine reading comprehension (Yu et al., 2018).

On the other hand, phrasal interpretation is an important aspect for many NLP tasks, and forms the basis of Phrase-Based Machine Translation (Koehn, 2010). Phrasal (n-gram based) alignments (Koehn et al., 2003) can model one-to-one, one-to-many, many-to-one, and many-to-many relations between target and source tokens while exploiting local context. They are also robust to non-compositional phrases. Despite the advantages, the concept of explicit phrasal attentions has largely been neglected in neural NLP. In fact, most language generation models produce sentences token-by-token autoregressively, and tend to use the token-based attention method which is order invariant.

Therefore, the intuition of phrasal alignments is vague in existing systems that solely depend on the underlying neural architectures (recurrent, convolutional, or self-attention) to incorporate contextual information. However, the information aggregation strategies employed by the underlying neural architectures provide context-relevant clues to represent only the current token, and do not explicitly model phrasal alignments. We argue that having an explicit inductive bias for phrases and phrasal alignments is beneficial for neural sequence transduction models to exploit the strong correlation between source and target language phrases. This would also make self-attention more capable of tackling other sequence modeling tasks such as language modeling.

In this paper, we propose a novel $n$-gram-level attention technique to leverage phrasal alignments in various NLP tasks. The technique is designed to assign attention scores directly to phrases in the source and compute phrase-level attention vectors for the target token. It is then applied in a new attention structure to conduct token-to-token and token-to-phrase mappings.

To show the effectiveness of our approach, we apply our phrase-based attention method to all multi-head attention layers of the Transformer network. Our experiments on WMT'14 translation tasks show improvements of up to 0.69, 1.84 and 1.65 BLEU in English-to-German, English-to-Russian and English-to-French translation tasks respectively, compared to the baseline Transformer trained in identical settings. Furthermore, when evaluated on the language modeling task, our phrasal models outperform the Transformer base model by up to 4.6 points in perplexity on the one-billion-word language modeling task. We make our code available at anonymous for research purposes.

## 2 BACKGROUND

Most NMT models adopt an encoder-decoder framework, where the encoder network first transforms an input sequence of symbols $x = (x_1, x_2, \ldots, x_n)$ to a sequence of continuous representations $Z = (z_1, z_2, \ldots, z_n)$. From this, the decoder generates a target sequence of symbols $y = (y_1, y_2, \ldots, y_m)$ autoregressively, one element at a time. Recurrent seq2seq models with diverse structures and complexity (Sutskever et al., 2014; Bahdanau et al., 2014; Luong et al., 2015; Wu et al., 2016) were the first to yield state-of-the-art results. Convolutional seq2seq models (Kalchbrenner et al., 2016; Gehring et al., 2017; Kaiser et al., 2018) alleviate the drawback of sequential computation of recurrent models and leverage parallel computation to reduce training time. Wu et al. (2019) recently proposed a light-weight convolution, which shows very promising results.

The Transformer network (Vaswani et al., 2017) structures the encoder and the decoder entirely with stacked self-attentions and cross-attentions (only in the decoder). In particular, it uses a multi-headed, scaled multiplicative attention defined as follows:

$$\text{Attention}(Q, K, V, W_q, W_k, W_v) = \text{softmax}\left(\frac{(QW_q)(KW_k)^T}{\sqrt{d_k}}\right)(VW_v) \tag{1}$$

$$\text{Head}^i = \text{Attention}(Q, K, V, W_q^i, W_k^i, W_v^i) \text{ for } i = 1 \ldots h \tag{2}$$

$$\text{AttentionOutput}(Q, K, V, W) = \text{concat}(\text{Head}^1, \text{Head}^2, \ldots, \text{Head}^h)W \tag{3}$$

where $Q \in \mathbb{R}^{l_q \times d}$, $K \in \mathbb{R}^{l_k \times d}$, and $V \in \mathbb{R}^{l_k \times d}$ are the matrices with query, key, and value vectors respectively, with $d$ being the number of dimensions; $W_q^i$, $W_k^i$, $W_v^i$ are the head-specific weights for query, key, and value vectors respectively, and $W$ is the weight matrix that combines the outputs of the heads. To encode a source sequence, the encoder applies self-attention, where $Q$, $K$ and $V$ identically contain the same vectors coming from the output of the previous layer. In the decoder, each decoder layer first applies the masked-self-attention over the outputs from the previous layer. The results are then used as queries to compute cross-attentions (encoder-decoder attentions) over the encoder states. For cross-attention, $Q$ comprises of the decoder self-attention states, while $K$ and $V$ comprise of the encoder states. We refer the reader to (Vaswani et al., 2017) for further details. Since the self-attention based encoder (or decoder) considers the input as a fully-connected directed graph, it can model long-range dependencies by explicitly attending to all tokens, and the maximum path length that signals need to travel is $O(1)$. The non-sequential computation also makes it highly parallelizable to multiple threads.

However, one crucial issue with the attention mechanisms employed in the Transformer as well as other NMT architectures is that they are order invariant locally and globally. That is, changing the order of the vectors in $Q$, $K$ and $V$ does not change the resulting attention weights and vectors. If this problem is not tackled properly, the model may not learn the sequential characteristics of the sentences. RNN-based models (Bahdanau et al., 2014; Luong et al., 2015) tackle this issue with a recurrent encoder and decoder, CNN-based models like (Gehring et al., 2017) use position embeddings, while the Transformer uses positional encoding. Shaw et al. (2018) further encode relative positions inside the self-attention layers.

Another limitation is that these attention methods attend to tokens, and play a role analogous to word alignment models in traditional SMT. It is, however, well admitted in SMT that phrases are better as translation units than words (Koehn, 2010). Without explicit attention to phrases, a particular attention function has to depend entirely on the token-level scores of a phrase for phrasal alignment, which is not robust or reliable, thus making it difficult for the model to learn the required mappings. For example, the attention heatmaps of the Transformer (Vaswani et al., 2017) show a concentration

of the scores on individual tokens even if it uses multiple heads concurrently in multiple layers. Our main hypothesis is that in order to exploit the strong correlation between source and target phrases, the NMT models should have explicit inductive biases for phrases.

There exists some research on phrase-based decoding in the NMT framework. For example, Huang et al. (2018) proposed a phrase-based decoding approach based on a soft reordering layer and a Sleep-WAke Network (SWAN), a segmentation-based sequence model proposed by Wang et al. (2017a). Their decoder uses a recurrent architecture without any attention on the source. Tang et al. (2016) and Wang et al. (2017b) used an external phrase memory to decode phrases for Chinese-to-English translation. In addition, hybrid beam search with phrase translation features from the statistical phrase table of a phrase-based SMT system was used to perform phrasal translation in (Dahlmann et al., 2017). Nevertheless, to the best of our knowledge, our work is the first to embed phrases into attention modules, which then propagate the information through the entire end-to-end Transformer network, including the encoder, decoder, and the cross-attention.

## 3 PHRASAL ATTENTION MODEL

In this section, we present our phrase-based attention model. In Subsection 3.1, we first describe our proposed module to compute attention weights and attention vectors based on $n$-grams of queries, keys, and values. Then, in Subsection 3.2, we describe how we use this module to compute different kinds of phrasal attentions. We describe our methods in the context of the Transformer (Vaswani et al., 2017). However, it is straight-forward to adopt them into other architectures such as the RNN- or CNN-based seq2seq models.

**Key Operations and Notations.** The core element in our method is a temporal (or one-dimensional) *convolutional operation* that is applied to a sequence of vectors representing the tokens in a sequence. Formally, we define the convolutional operator applied to each token $x_i$ with the corresponding vector representation $\mathbf{x}_i \in \mathbb{R}^d$ as:

$$a_{i,j} = \mathbf{w}_j^T (\oplus_{k=0}^{n-1} \mathbf{x}_{i+k}) \tag{4}$$

where $n$ is the convolution window size, $\mathbf{w}_j \in \mathbb{R}^{nd}$ is the kernel weight taken from the weight matrix $\boldsymbol{W} \in \mathbb{R}^{n \times d \times d}$, and $\oplus$ denotes a vector concatenation that produces a vector in $\mathbb{R}^{nd}$. The convolution over the entire sequence $\boldsymbol{X} \in \mathbb{R}^{l \times d}$ of length $l$ gives a feature map $\mathbf{a}_j$. By repeating this process with $d$ different weight vectors from $\boldsymbol{W}$, we get a $d$-dimensional representation for each token $x_i$. In the rest of the paper, we will use the following notations.

- $\text{Conv}_n(\boldsymbol{X}, \boldsymbol{W})$ to denote a convolution operation with window size $n$ and kernel weights $\boldsymbol{W} \in \mathbb{R}^{n \times d \times d}$ over an input sequence $\boldsymbol{X} \in \mathbb{R}^{l \times d}$, based on the definition in Equation 4. The result of this convolution is a matrix in $\mathbb{R}^{(l-n+1) \times d}$, whose columns represent the feature maps corresponding to the kernels, and rows represent feature representations corresponding to the tokens in the sequence.

- $\text{SConv}_n(\boldsymbol{X}, \mathbf{Y})$ to denote a *serial* convolution operation with window size $n$ over an input $\boldsymbol{X} \in \mathbb{R}^{l \times d}$ for all the *rows* $\boldsymbol{W}_i \in \mathbb{R}^{n \times d \times 1}$ in the kernel *tensor* $\mathbf{Y} = (\boldsymbol{W}_1, \dots, \boldsymbol{W}_t) \in \mathbb{R}^{t \times n \times d \times 1}$. In other words, $\text{SConv}_n(\boldsymbol{X}, \mathbf{Y}) = \oplus_{i=0}^{t-1} \text{Conv}_n(\boldsymbol{X}, \boldsymbol{W}_i)$ that produces a matrix in $\mathbb{R}^{t \times (l-n+1)}$.

- $\rho_n(\boldsymbol{x})$ to denote a *reshape*[1] operation on the vector $\boldsymbol{x} \in \mathbb{R}^{nd}$ to reshape it to a matrix $\boldsymbol{X} \in \mathbb{R}^{n \times d \times 1}$, and $\psi_n(\boldsymbol{X})$ to denote a *serial reshape* operation to reshape the matrix $\boldsymbol{X} \in \mathbb{R}^{t \times (nd)}$ to a tensor $\mathbf{X} \in \mathbb{R}^{t \times n \times d \times 1}$.

- $\text{softmax}(\boldsymbol{A})$ to denote a *softmax* operation over each row of the matrix $\boldsymbol{A}$. Formally, for $\boldsymbol{A}' = \text{softmax}(\boldsymbol{A})$, each entry $A'(i,j)$ is computed as: $A'(i,j) = \frac{\exp(A(i,j))}{\sum_j \exp(A(i,j))}$.

---

[1] *Tensorflow:* tf.reshape(); *Pytorch:* .view()

### 3.1 $n$-GRAM-LEVEL ATTENTION

Our $n$-gram based attention method computes the attention weights and vectors based on $n$-gram representations of the tokens. In particular, when computing the attention scores $\boldsymbol{a}_i \in \mathbb{R}^{l_k}$ for each query $\boldsymbol{q}_i$ in $\boldsymbol{Q} = (\boldsymbol{q}_1, \ldots, \boldsymbol{q}_{l_q}) \in \mathbb{R}^{l_q \times d}$, we use the query vector $\boldsymbol{q}_i$ as the kernel in the convolution operation applied to the set of keys $\boldsymbol{K} = (\boldsymbol{k}_1, \ldots, \boldsymbol{k}_{l_k}) \in \mathbb{R}^{l_k \times d}$. More formally,

$$\boldsymbol{a}_i = \frac{\mathrm{Conv}_n(\boldsymbol{K}\boldsymbol{W}_k, \rho_n(\mathbf{q}_i^T \boldsymbol{W}_q))}{\sqrt{d * n}} \tag{5}$$

where $\boldsymbol{W}_k \in \mathbb{R}^{d \times d}$ and $\boldsymbol{W}_q \in \mathbb{R}^{d \times (nd)}$ are trainable weights (head weights) to linearly transform $\boldsymbol{K}$ and $\boldsymbol{Q}$ respectively, and $\rho_n$ is the required reshaping operation to make the dimensions compatible. Applying Equation 5 for all the queries in $\boldsymbol{Q}$ gives the score matrix $\boldsymbol{A} = (\boldsymbol{a}_1, \ldots, \boldsymbol{a}_{l_q}) \in \mathbb{R}^{l_q \times l_k}$, where each row vector $\mathbf{a}_i \in \boldsymbol{A}$ contains the attention scores for the query vector $\mathbf{q}_i \in \boldsymbol{Q}$.

Next, we compute the attention (or context) vectors as follows.

$$\boldsymbol{C} = \mathrm{softmax}(\boldsymbol{A}) \, \mathrm{Conv}_n(\boldsymbol{V}, \boldsymbol{W}_v) \tag{6}$$

where $\boldsymbol{W}_v \in \mathbb{R}^{n \times d \times d}$ are the kernel parameters to achieve the $n$-gram representations by convolving over the value vectors $\boldsymbol{V} = (\boldsymbol{v}_1, \ldots, \boldsymbol{v}_{l_k}) \in \mathbb{R}^{l_k \times d}$, which are in turn linearly combined using the attention weights computed by the softmax.

By using the queries as the kernel parameters, we allow the queries to *dynamically* (kernels vary for different queries) and *directly* interact with the window of key vectors and compute the $n$-gram based attention scores. An alternative way to achieve phrasal attentions would be to use $\boldsymbol{A} = \frac{(\boldsymbol{Q}\boldsymbol{W}_q) \, \mathrm{Conv}_n(\boldsymbol{K}, \boldsymbol{W}_k)^T}{\sqrt{d}}$ to compute the attention scores in Equation 5. In contrast to our approach, this method is *static* and *indirect* in the sense that the convolution uses a static kernel and the queries do not interact directly with the keys; instead the model relies on the kernel weights ($\boldsymbol{W}_k$) to learn $n$-gram patterns. In our initial experiments, we also found the direct approach to perform better.

### 3.2 MULTI-HEADED PHRASAL ATTENTION

Having presented our core method to perform attentions based on $n$-grams, we now introduce our novel extension to the multi-headed attention framework of the Transformer to enable it to pay attention not only to tokens but also to other $n$-grams across many sub-spaces and locations. In particular, each head uses the token (unigram) representation of the query to attend to all $n$-gram types (*e.g.,* $n = 1, 2, \ldots, N$) simultaneously. To achieve this, we first use the query vectors to compute the attention scores for each $n$-gram type separately by performing convolution over the key vectors with the respective window sizes. All the $n$-gram scores are then concatenated before passing them through a *softmax* to compute the attention weights over all $n$-grams. Similarly, the value vectors for the $n$-gram types are concatenated to produce the overall attention output. Figure 1 exemplifies the process for attentions over unigrams and bigrams. For self-attention, the query vectors in $\boldsymbol{Q}$ represent the unigrams ('India', 'and', 'Japan') of the input sequence, whereas for cross-attention, they represent the decoder states of the target side tokens. The overall attention output $\boldsymbol{C}_n$ can be formally described by the following sequence of equations:

$$\boldsymbol{Q}_1 = \boldsymbol{Q}\boldsymbol{W}_{q,1}; \quad \boldsymbol{Q}_n = \boldsymbol{Q}\boldsymbol{W}_{q,n} \tag{7}$$

$$\boldsymbol{A}_1 = \frac{\boldsymbol{Q}_1(\boldsymbol{K}\boldsymbol{W}_k)^T}{\sqrt{d}} \tag{8}$$

$$\boldsymbol{A}_n = \frac{\mathrm{SConv}_n(\boldsymbol{K}\boldsymbol{W}_k, \psi_n(\boldsymbol{Q}_n))}{\sqrt{d * n}} \tag{9}$$

$$\boldsymbol{S}_n = \mathrm{softmax}([\boldsymbol{A}_1; \boldsymbol{A}_2; \ldots; \boldsymbol{A}_n]) \tag{10}$$

$$\boldsymbol{V}_n = [(\boldsymbol{V}\boldsymbol{W}_{v,1}); \mathrm{Conv}_2(\boldsymbol{V}, \boldsymbol{W}_{v,2}); \ldots; \mathrm{Conv}_n(\boldsymbol{V}, \boldsymbol{W}_{v,n})] \tag{11}$$

$$\boldsymbol{C}_n = \boldsymbol{S}_n\boldsymbol{V}_n \tag{12}$$

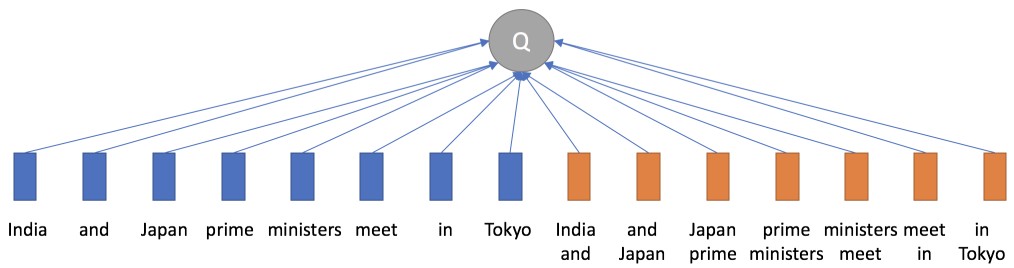

Figure 1: Multi-Headed phrasal attention for each head. Each query token attends to uni- and bi-grams of key/value tokens.

where $\boldsymbol{W}_{q,n} \in \mathbb{R}^{d \times (nd)}$ and $\boldsymbol{W}_{v,n} \in \mathbb{R}^{n \times d \times d}$ are the respective weight matrices for the queries and values, and $\boldsymbol{W}_k \in \mathbb{R}^{d \times d}$ is a weight matrix for the keys that is shared across all $n$-gram types.

Note that the *serial convolution* $\mathrm{SConv}_n()$ and the *serial reshape* $\psi_n()$ functions in Equation 9 implies the computation of all attention scores $\mathbf{a}_i \in \boldsymbol{A}_n$ (*i.e.,* applying $\mathrm{Conv}_n$ and $\rho_n$ in Equation 5 for each row $\mathbf{q}_i$ of $\boldsymbol{Q}$). We can aggregate multiple $n$-gram types within an attention module (*e.g.,* 1-2-3 grams). In our method, we do not need to pad the input sequences before the convolution operation to ensure identical sequence length. The key/value sequences that are shorter than the window size do not have any valid phrasal component to be attended.

## 4 EXPERIMENTS

To demonstrate the effectiveness of our phrase-based attention method, we experiment with two different tasks: machine translation (MT) and language modeling (LM). In the following, we present the training settings, experimental results and analysis of our models on these two tasks.

### 4.1 TRAINING SETTINGS

We replicate most of the training settings from (Vaswani et al., 2017) for our models, to enable a fair comparison with the original Transformer and the Transformer with relative positioning (Shaw et al., 2018). Specifically, we use the Adam optimizer (Kingma & Ba, 2014) with $\beta_1 = 0.9$, $\beta_2 = 0.98$, and $\epsilon = 10^{-9}$. We follow a similar learning rate schedule with a $warmup\_steps$ of 16000 updates: $LearningRate = 2 \times d^{-0.5} \times \min(step\_num^{-0.5}, step\_num \times warmup\_steps^{-1.5})$.

Similar to Vaswani et al. (2017), we also applied residual dropout with 0.1 probability and label smoothing with $\epsilon_{ls} = 0.1$. Our models are implemented in the *tensor2tensor*[2] library (Vaswani et al., 2018), on top of the original Transformer codebase.

We conducted all the experiments with our models and the original Transformer in an identical setup for a fair comparison. While Vaswani et al. (2017) trained their base and big models at a massive scale with 8 GPUs, we could only train our models and the baselines on a single GPU because of limited GPU facilities. However, we could (virtually) replicate the 8-GPU setup with a single GPU following the *gradient aggregation* method proposed recently by Ott et al. (2018).

**Setup for MT Experiments.** To compare our models with state-of-the-art models, we train all the models with the identical 8-GPU (by gradient aggregation) setup on WMT'16 English-German (En-De), WMT'17 English-Russian (En-Ru) and WMT'14 English-French (En-Fr) datasets. The training datasets contain about 4.5, 25, and 35 million sentence pairs for En-De, En-Ru, and En-Fr, respectively. The effective batches were formed by sentence pairs containing approximately 32,768 source and 32,768 target tokens.

All translation tasks are evaluated in case-sensitive tokenized BLEU. For validation (development) purposes, we use newstest2013 for En-De, newstest2016 for En-Ru, and a random split from the

---

[2]https://github.com/tensorflow/tensor2tensor

| Model | #-Params | N-grams | En→De | En→Ru | En→Fr |
|---|---|---|---|---|---|
| **Base size** | | | | | |
| Vaswani et al. (2017) | 63M | - | 27.16 | 34.37 | 39.21 |
| Shaw et al. (2018) | 63M | - | 27.20 | 33.59 | 39.37 |
| **Big size** | | | | | |
| Vaswani et al. (2017) | 214M | - | 27.77 | 35.54 | 40.61 |
| **Ours, Base size** | | | | | |
| Ours | 81M | 1-2 | 27.70 | 35.44 | 40.57 |
| Ours | 110M | 1-2-3 | **27.85** | **36.21** | **40.86** |

Table 1: BLEU (cased) scores on WMT'14 testsets for En→De, En→Ru, and En→Fr translation tasks. All the models were trained with gradient aggregation to replicate a **8-GPU** setup on a single physical GPU.

training set for En-Fr. We evaluate all our models on WMT'14 translation tasks (newstest2014 test sets). We use Byte-Pair Encoding or BPE (Sennrich et al., 2016) with a combined (source and target) vocabulary of 37,000 subwords for En-De, 40,000 subwords for En-Ru and En-Fr. We take the average of the last 5 checkpoints (saved at 5000-update intervals) for evaluation, and use a beam search size of 5 and length penalty of 0.6 (Wu et al., 2016).

**Setup for LM Experiments.** For our LM experiments, we use the One Billion Word Benchmark dataset (Chelba et al., 2013), which contains 768 million words of data compiled from WMT 2011 News Crawl data,[3] with a vocabulary of 32,000 words. We use its held-out data as the test set. We train the base-size models (monolingual decoders) on virtually 4 GPUs for 100,000 *updates* by using the gradient aggregation technique on a single GPU. The effective batch size is 16,384 tokens.

## 4.2 MACHINE TRANSLATION RESULTS

**Comparison with State-of-the-art.** In Table 1, we compare our models on the En→De, En→Ru and En→Fr translation tasks with the base and big size Transformer models (Vaswani et al., 2017), and the base size Transformer with relative position model (Shaw et al., 2018). All the models are trained in the identical **8-GPU setup** using gradient aggregation for 100K updates for En→De, 120K updates for En→Ru, and 150K updates for En→Fr.

Generally, we can see that our models outperform the base models. Some of them also surpass the Transformer big model trained in identical settings, while having only half the number of parameters.

More specifically, on the En→De translation task, our base-size model with 1-2 grams achieves a BLEU of 27.70, exceeding the Transformer base by 0.6 BLEU. Including higher order $n$-grams contributes further improvements, reaching up to 27.85 BLEU for 1-2-3 grams. It also performs on par with the big Transformer, while requiring less than half of the number of parameters.

Likewise, on the En→Ru translation task, our models boost the performances significantly over the Transformer base. They achieve 36.21 BLEU with 1-2-3 grams, which is 1.84 points higher than that of the Transformer base. They also outperform the Transformer big by 0.67 points.

Similar trends can also be seen in the En→Fr translation task, where our model with 1-2-3 grams scores at 40.86 BLEU, exceeding the Transformer base by 1.65 points. It also outdoes the Transformer big by 0.2, even though it has considerably fewer parameters. These results demonstrate the effectiveness of our approach over the existing methods.

**Effect of Higher-Order $n$-grams and Training Batch Size.** To analyze our models further with respect to the effect of higher-order $n$-grams and the impact of training batch size, we conducted another set of experiments on En-De and En-Ru translation tasks (both directions) in a **single-GPU setup**, and compared with the Transformer base and big models on the same setup. The effective

---

[3]http://www.statmt.org/lm-benchmark/

| Model | N-grams | En→De | De→En | En→Ru | Ru→En |
|---|---|---|---|---|---|
| Transformer big | - | 26.62 | —— | 32.31 | —— |
| Transformer base | - | 26.31 | 29.76 | 33.12 | 32.87 |
| Ours | 1-2 | 27.11 | 30.16 | 34.18 | 33.12 |
| Ours | 1-2-3 | 27.37 | **30.55** | 34.72 | 33.53 |
| Ours | 1-2-3-4 | **27.37** | 30.04 | **34.90** | **33.70** |

Table 2: BLEU (cased) scores on WMT'14 testsets for English-German (En-De) and English-Russian (En-Ru) language pairs (in both directions). All models were trained with **1 GPU** (no gradient aggregation). The decrease in scores in this table compared to the ones in Table 1 is due to the number of GPUs used (1 vs. 8).

| Model | N-grams | Perplexity |
|---|---|---|
| Vaswani et al. (2017) | - | 46.37 |
| Shaw et al. (2018) | - | 46.13 |
| Ours | 1-2 | 41.77 |

Table 3: Perplexity scores on one-billion-word language modeling benchmark. All models are of **base-size** and were trained for 100K updates with gradient aggregation to produce a virtual 4-GPU setup on a single GPU.

batch size in this experiment was 4096 tokens. En-De models were trained for 500K updates while En-Ru ones were trained for 900K updates. The results are shown in Table 2.

It is evident that almost all of our models achieve higher BLEU scores than the Transformer base and big models. For the En-De pair, our model with 1-2 grams achieves 27.11 and 30.16 BLEU for En→De and De→En translation tasks, giving improvements of 0.8 and 0.4 points, respectively. Including higher-order $n$-grams does improve the performance (for 1-2-3 grams), but diminishing patterns are also observed (for 1-2-3-4 grams) in these tasks.

On the other hand, for the En-Ru pair, the models with higher-order $n$-grams excel in BLEU performances. Specifically, with 1-2-3-4 grams it yields up to 34.90 and 33.70 BLEU for En→Ru and Ru→En, surpassing the baselines by 1.78 and 0.83 points, respectively.

If we compare the corresponding translation results for En→De and En→Ru in Table 1 vs. Table 2, we notice a significant drop in BLEU for the 1-GPU setup across all the models, though the 1-GPU models were offset by longer training (more updates). The performance of the Transformer big is outstandingly low in the 1-GPU setup. There has been evidence that practical training of the Transformer (theirs and ours) is significantly susceptible to the batch size (which increases with the number of GPUs used), and training on a single GPU with a lower batch size for sufficiently long does not produce similar results as with 8 or more GPU settings; please see Popel & Bojar (2018); Ott et al. (2018) and the discussions[4] for details on this issue.

## 4.3 LANGUAGE MODELING RESULTS

Table 3 presents the results on the language modeling task, where we compare our model with uni- and bi-gram attentions against the three baseline models in terms of perplexity. Our model outperforms the Transformer base by 4.6 perplexity points. It also surpasses the Transformer with relative positional encoding (Shaw et al., 2018) by a comparable margin. These results demonstrate that our models are superior to the baselines in the language modeling task, as well.

## 4.4 MODEL INTERPRETATION

To interpret our phrasal attention models, we now discuss how they learn the alignments. Figure 2 and 3 show attention heatmaps for an En→De sample in newstest2014; Figure 2 displays the heatmap in layer 3 (mid layer), while Figure 3 shows the one in layer 6 (top layer) within a 6-layer

---

[4]https://github.com/tensorflow/tensor2tensor/issues/444

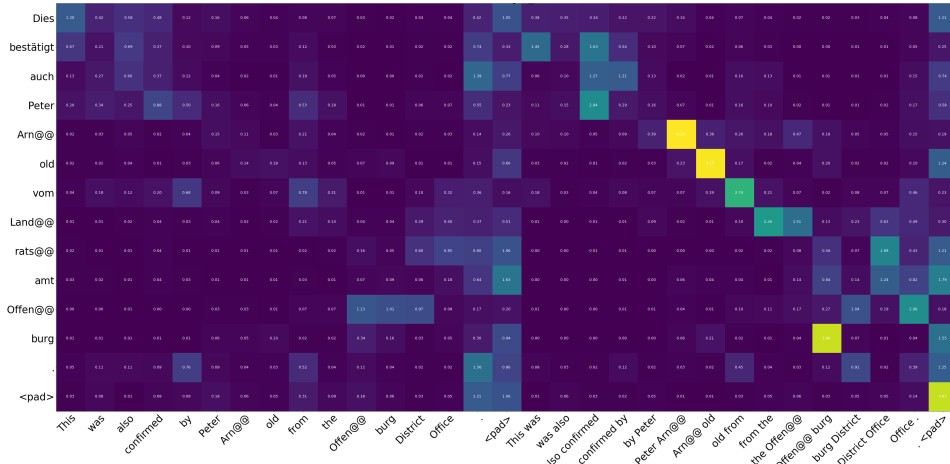

Figure 2: Attention heat maps at layer 3 of our model for a sample sentence pair in English-German newstest2014 test set. The left half in each figure indicates token-to-token mappings, while the right half indicates token-to-phrase mappings.

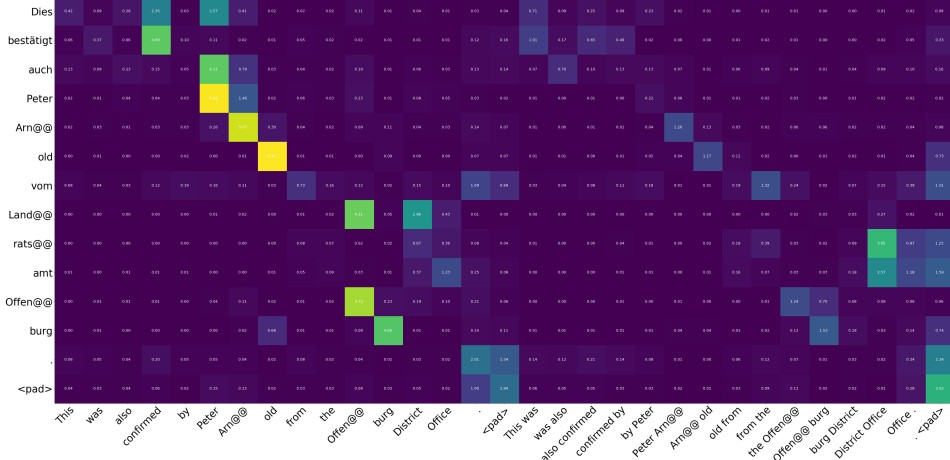

Figure 3: Attention heat maps at layer 6 of our model for a sample sentence pair in English-German newstest2014 test set. The left half in each figure indicates token-to-token mappings, while the right half indicates token-to-phrase mappings.

Transformer with our phrasal attention. Each figure shows two squares representing token-to-token (left square) and token-to-phrase (right square) attentions, respectively. We can see in the two figures that phrasal (token-to-phrase) attentions are activated strongly in the mid-layers. On the other hand, token-token attentions are activated the most in the top layer, whose final representations are used to predict translated tokens. Although the distribution of attentions can vary depending on model initialization, we observed that 50%-60% of the attentions are concentrated on phrases.

## 5 CONCLUSION

We have presented novel approaches to incorporating phrasal alignments into the attention mechanism of state-of-the-art sequence transduction models. Our methods assign attentions to both tokens and phrases of the source sequences. While we have applied our attention mechanism to the Transformer network, it is generic and can be implemented in other architectures. We have shown the effectiveness of our approach on two NLP tasks. On machine translation, our models show significant gains on WMT'14 English-German, English-Russian, and English-French translation tasks. Our phrasal models also outperform the Transformer in the one-billion-word language modeling task. We are planning future extensions of our techniques to other tasks, such as summarization and question answering. We also plan to improve our models with a phrase-based decoding procedure.

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
