# OpenReview forum: "Enhancing Attention with Explicit Phrasal Alignments"
_ICLR.cc/2020/Conference — Reject_

### Official Review · AnonReviewer3 · 2019-10-23
**Official Blind Review #3**

**Rating:** 6

**Review:**

This paper proposes an extension of the attention module that explicitly incorporates phrase information. Using convolution, attention scores are obtained independently for each n-gram type, and then combined. Transformer models with the proposed phrase attention are evaluated on multiple translation tasks, as well as on language modelling, generally obtaining better results than by simply increasing model size.

I lean towards the acceptance of the paper. The approach is fairly well motivated, likely easy to implement and results are mostly convincing. However, some claims may be too strong and I had difficulty understanding some parts of the approach.

I find the idea interesting. Standard attention is unbiased to distance (but sensitive to it because of positional embeddings). Phrasal attention may be a useful learning bias, giving particular importance to nearby words.

On 3 WMT'14 translation tasks, the proposed approach leads to improvements between 0.7 and 1.8 BLEU with respect to Transformer Base. Running each model with different random seeds and presenting statistical significance results would be ideal, but such runs can be expensive given the size of the datasets. Using phrasal attention appears to be more efficient than simply increasing model size. In addition to the number of parameters, the number of FLOPs per update might also be useful to know. Phrasal attention also leads to lower perplexity on a large-scale modeling task, although I can't confidently evaluate the importance of this result.

While results in the model interpretation section are cherry-picked, they illustrate that the model can use the additional capacity provided by phrasal attention. There is also clear qualitative differences between layers.

Some equations are confusing. For example, in Eq. 5, the right-most argument of Conv_n() appears to be of dimension nxdx1, but convolutions are defined for dimension nxdxd. I would suggest going over the presentation of phrasal attention carefully (or correct me if I interpreted the notation wrongly).

Some claims made in the paper may be too strong. While there are similarities between alignment and attention, they are not necessarily interchangeable in neural models. For example, (Koehn and Knowles. Six Challenges for Neural Machine Translation) show that they can be mismatched (Fig. 9).

Moreover, while input embeddings (and arguably the last decoder hidden layer) mostly contain token-level information, intermediate representations merge information from multiple positions. As such, at a given layer, it is not guaranteed the the i^{th} vector is a representation i^{th} token. For example, (Voita et al. The Bottom-up Evolution of Representations in the Transformer: A Study with Machine Translation and Language Modeling Objectives) show that the mutual information between an input token and its corresponding encoder representation diminishes as depth increases. As such, neighbouring representations may not represent n-grams.

It would be appropriate to compare the proposed approach to (Hao et al. Multi-Granularity Self-Attention for Neural Machine Translation). However, this is very recent work (September 5 on ArXiv), so it would be understandable for the authors not to know about it.

Questions:

While searching for related work, I found an earlier submitted version of this paper ("Phrase-Based Attentions", submitted to ICLR 2019). The reported numbers differ from the current version. Why?

**Experience Assessment:**

I have read many papers in this area.

**Review Assessment: Checking Correctness Of Derivations And Theory:**

I assessed the sensibility of the derivations and theory.

**Review Assessment: Checking Correctness Of Experiments:**

I assessed the sensibility of the experiments.

**Review Assessment: Thoroughness In Paper Reading:**

I read the paper at least twice and used my best judgement in assessing the paper.

---

### Official Review · AnonReviewer1 · 2019-10-23
**Official Blind Review #1**

**Rating:** 3

**Review:**

This work aims to incorporate phrase representation into attention mechanism. The proposed method is straightforward (which is good): a convolution window with size n is used to calculate representation for an n-gram, which then replaces the token representation in a standard attention model. The paper implements the multihead version of the proposed phrase-based attention, more specifically, in a transformer model. Experiments with machine translation and language modeling show that it outperforms the token attention counterpart.

My main concerns are about the experiments:

- For the translation experiments, the transformer numbers are lower than those reported by Vaswani et al. (2017) across the board, for both the "base" and the "big" settings. I didn't find convincing reason for this. Could the authors comment on this, and also on why they do not directly compare against Vaswani et al. (2017). The same for the language modeling experiments.

- Table 3. To the best of my knowledge, neither Vaswani et al. (2017) or Shaw et al. (2018) report language modeling results. I'm guessing the authors use their implementation and establish their own baselines, which is okay. But please indicate this instead of putting a citation in the table, which can be misleading.

- Minor: the start of Section 3 reads like source code and comment, and is a bit hard for me to follow.

I do not recommend the that the paper is accepted, until the authors address my concerns on the baselines.


Missing references:
PaLM: A Hybrid Parser and Language Model. https://arxiv.org/abs/1909.02134

**Experience Assessment:**

I have published in this field for several years.

**Review Assessment: Checking Correctness Of Derivations And Theory:**

N/A

**Review Assessment: Checking Correctness Of Experiments:**

I assessed the sensibility of the experiments.

**Review Assessment: Thoroughness In Paper Reading:**

I read the paper at least twice and used my best judgement in assessing the paper.

---

### Official Review · AnonReviewer2 · 2019-10-24
**Official Blind Review #2**

**Rating:** 8

**Review:**

This work proposes an attention mechanism that directly reflects the phrasal correspondence by employing convolution. The n-gram aware attention is incorporated into Transformer and shows gains in translation and language modeling tasks.

It is a novel and straightforward way to incorporate phrasal relation in the attention mechanism. The gains reported in this paper are meaningful, thought might not be SOTA. The analysis on attention is also interesting in that the phrasal relation is employed in lower layers, but not in the higher layer.

Other comment:

- It is not clear to me how the n-gram aware phrasal attention is incorporated into multi-headed attention described in section 3.2. Did you completely remove the multi-head attention, but used only n-gram attentions? Or, did you keep multi-head attention and incorporated phrasal attention for each head?

- It is unfortunate that this work does not report empirical results when applied to a big model configuration. Did you encounter OOM when running on a big model?

**Experience Assessment:**

I have published in this field for several years.

**Review Assessment: Checking Correctness Of Derivations And Theory:**

I carefully checked the derivations and theory.

**Review Assessment: Checking Correctness Of Experiments:**

I assessed the sensibility of the experiments.

**Review Assessment: Thoroughness In Paper Reading:**

I read the paper at least twice and used my best judgement in assessing the paper.

---

### Decision · Program_Chairs · 2019-12-19

**Decision:**

Reject

**Comment:**

This paper proposes a phrase-based attention method to model word n-grams (as opposed to single words) as the basic attention units. Multi-headed phrasal attentions are designed within the Transformer architecture to perform token-to-token and token-to-phrase mappings. Some improvements are shown in English-German, English-Russian and English-French translation tasks on the standard WMT'14 test set, and on the one-billion-word language modeling benchmark.

While the proposed approach is interesting and takes inspiration in the notion of phrases used in phrase-based machine translation, with some positive empirical results, the technical novelty of this paper is rather limited, and the experiments could be more solid. While it is understandable that lack of computational resources made it hard to experiment with larger models (e.g. Transformer-big), perhaps it would be interesting to try on language pairs with fewer resources (smaller datasets), where base models are more competitive.